# The Update Immune-Regulatory Role of Pro- and Anti-Inflammatory Cytokines in Recurrent Pregnancy Losses

**DOI:** 10.3390/ijms24010132

**Published:** 2022-12-21

**Authors:** Xiuhua Yang, Yingying Tian, Linlin Zheng, Thanh Luu, Joanne Kwak-Kim

**Affiliations:** 1Department of Obstetrics, The First Hospital of China Medical University, Shenyang 110001, China; 2The First Hospital of China Medical University, Shenyang 110001, China; 3Reproductive Medicine and Immunology, Obstetrics and Gynecology, Clinical Sciences Department, Chicago Medical School, Rosalind Franklin University of Medicine and Science, Vernon Hills, IL 60061, USA; 4Clinical Immunology Laboratory, Foundational Sciences and Humanities, Microbiology and Immunology, Chicago Medical School, Rosalind Franklin University of Medicine and Science, North Chicago, IL 60064, USA

**Keywords:** recurrent pregnancy losses, cytokine, pregnancy, immune, miscarriage

## Abstract

Recurrent pregnancy losses (RPL) is a common reproductive disorder with various underlying etiologies. In recent years, rapid progress has been made in exploring the immunological mechanisms for RPL. A propensity toward Th2 over Th1 and regulatory T (Treg) over Th17 immune responses may be advantageous for reproductive success. In women with RPL and animals prone to abortion, an inordinate expression of cytokines associated with implantation and early embryo development is present in the endometrium or decidua secreted from immune and non-immune cells. Hence, an adverse cytokine milieu at the maternal-fetal interface assaults immunological tolerance, leading to fetal rejection. Similar to T cells, NK cells can be categorized based on the characteristics of cytokines they secrete. Decidual NK (dNK) cells of RPL patients exhibited an increased NK1/NK2 ratio (IFN-γ/IL-4 producing NK cell ratios), leading to pro-inflammatory cytokine milieu and increased NK cell cytotoxicity. Genetic polymorphism may be the underlying etiologies for Th1 and Th17 propensity since it alters cytokine production. In addition, various hormones participate in cytokine regulations, including progesterone and estrogen, controlling cytokine balance in favor of the Th2 type. Consequently, the intricate regulation of cytokines and hormones may prevent the RPL of immune etiologies. Local or systemic administration of cytokines or their antagonists might help maintain adequate cytokine milieu, favoring Th2 over Th1 response or Treg over Th17 immune response in women with RPL. Herein, we provided an updated comprehensive review regarding the immune-regulatory role of pro- and anti-inflammatory cytokines in RPL. Understanding the roles of cytokines involved in RPL might significantly advance the early diagnosis, monitoring, and treatment of RPL.

## 1. Introduction

Recurrent pregnancy losses (RPL) is a common reproductive disorder affecting 2–5% of reproductive-age women [1], which is defined as two or more pregnancy losses before 20 weeks of pregnancy by the American Society for Reproductive Medicine [2]. RPL can be caused by embryonic chromosomal, immunological, anatomical, environmental, and thrombotic abnormalities [3,4]. Approximately 50% of RPL patients have unknown causes, called unexplained RPL (uRPL) [5]. Rapid progress has been made in understanding the immunological mechanisms for RPL [6]. Maternal immune abnormalities are often present in women with uRPL [7,8], such as an increased natural killer (NK) cell count and/or activity [9,10] and a tilted T helper (Th)1 over Th2 immune response, which is thought to be harmful to pregnancy [11]. The classification of T cell subsets has extended beyond Th1 and Th2 cells to more complex regulatory T cell (Treg) and Th17 cell subsets [12]. Women with RPL have the propensity to Th17 over Treg cell immunity [13]. Th17 cells produce interleukin (IL)-17, a pro-inflammatory cytokine that promotes inflammation and maternal-fetal rejection and interacts with Th1 cells, further contributing to the immunopathology of RPL [14]. Contrarily, Treg cells mediate maternal-fetal tolerance [15], allowing an embryo, a semi-allograft, to live without rejection in the uterus [7].

Regulatory uterine NK (uNK) cells refer to highly granulated NK cells located in the endometrium with the phenotype of CD56^++^/CD16^−^ [16]. They have the ability to secrete a range of cytokines [16]. In normal pregnancy, increased regulatory uNK cells are pivotal for maintaining reproductive success since they play a fundamental role in trophoblast invasion, spiral artery remodeling, and appropriate placentation [17,18,19]. Decreased number or dysfunction of regulatory uNK cells is closely related to RPL [20].

The interaction between tolerogenic dendritic cells (DCs) and Treg cells in pregnancy is pivotal for sustaining maternal-fetal immune tolerance. Tolerogenic DCs promote the proliferation and differentiation of Treg cells by increasing the expression of IL-10 [21]. These special Treg subsets play a crucial role in avoiding maternal immune response against embryo-paternal derived antigens. Furthermore, tolerogenic DCs interacting with trophoblasts could elevate the proportion of Tregs FoxP3^+^ [22]. However, this phenomenon does not exist in DCs that do not interact with trophoblast cells [22]. Similarly, NK cells express killer cell immunoglobulin-like receptors (KIRs), which interact with HLA-C derived from extravillous trophoblast cells (EVTs). The improper combination between KIR and HLA-C affects the ability of NK cells to secrete cytokines, including granulocyte-macrophage colony-stimulating factor (GM-CSF), thus participating in the occurrence of RPL [4].

Various cytokines play significant roles in the maintenance of pregnancy. The ability of embryos to avoid immune rejection is partially facilitated by the presence of fundamental cytokines in the peripheral blood and/or at the maternal-fetal interface. Therefore, cytokine gene polymorphisms, which modify cytokine production levels and functions, may contribute to RPL. Cytokines are classified as pro-inflammatory (Th1) or anti-inflammatory (Th2), depending on their functions. Th1 immunity induces a cell-mediated cytotoxic response to intracellular pathogens [7], whereas Th2 immunity primarily generates humoral immunity and immune tolerance [23]. Th1 pro-inflammatory cytokines are mostly tumor necrosis factor (TNF)-α, interferon (IFN)-γ, and IL-1β; Th2 cytokines are predominantly IL-4, IL-5, IL-10, and IL-13 [24,25]. During the implantation window, pro-inflammatory cytokines promote the invasion of trophoblast cells and endometrial neovascularization. However, prolonged or overexposure to pro-inflammatory cytokines may harm the pregnancy, resulting in miscarriage.

The endometrium secretes copious amounts of cytokines throughout the follicular and luteal phases of the menstrual cycle [26], contributing to a favorable uterine milieu for the preparation of implantation and placental development. The CD4^+^ T lymphocytes are the primary source of cytokines. Additionally, NK cells, macrophages, epithelial cells, mesenchymal cells, and other leucocytes produce cytokines at the maternal-fetal interface [27]. During trophoblast implantation, pregnancy hormones and direct stimulation by trophoblast cells improve the endometrial receptivity while the endometrium is gradually transformed into the decidua. Cytokines and receptors on decidual cells are primarily responsible for interacting with trophoblast cells. Cytokines have multiple functions, including regulation of embryo implantation, placental development, cytotrophoblast proliferation, vascular remodeling, trophoblast invasion, cell death, and the induction of embryo tolerance in the uterus [28,29].

Although the contribution of cytokines to RPL has been widely studied [30,31,32], their precise roles need further investigation. This review analyzes the updated immune-regulatory role of pro- and anti-inflammatory cytokines in maternal peripheral blood and endometrium, including Th1/Th2, Treg/Th17, and NK cell-related cytokines. Although other cells also produce cytokines at the maternal-fetal interface, such as trophoblasts, stromal cells, and macrophages, they are not within the scope of this review. Understanding the roles of cytokines involved in RPL might significantly advance the early diagnosis, monitoring, and treatment of RPL. This review highlights relevant studies with significant discoveries that merit additional investigation. Two independent scholars conducted a comprehensive publication search to collect the articles with the following keywords: “recurrent miscarriage,” “recurrent pregnancy losses,” “recurrent spontaneous abortion,” and “cytokine” from PubMed, Web of Science, and Google Scholar database. All articles were published in English from January 1985 to September 2022. Letters were not included.

## 2. Pro- and Anti-Inflammatory Cytokines in Maternal Peripheral Blood

According to a prior animal study exploring the Th1 immune response associated with RPL, trophoblast antigen could activate lymphocytes from abortion-prone female mice to produce pro-inflammatory cytokines, such as TNF-α [33]. The ratios of Th1/Th2 cytokine-producing T helper cells from RPL patients (IFN-γ/IL-4, IFN-γ/IL-10, TNF-α/IL-4 and TNF-α/IL-10 producing Th cell ratios) were considerably higher than normal pregnant women after in vitro stimulation of peripheral blood mononuclear cells (PBMCs) [30]. Another study investigated the Th1/Th2 cytokine ratios in the peripheral blood of uRPL women (*n* = 44) and normal pregnant women (*n* = 42). They reported that uRPL women had a higher IFN-γ/IL-4 cytokine ratio [34]. According to this study, the IFN-γ/IL-4 cytokine ratio had a sensitivity of 84.09% and a specificity of 69.05% for diagnosing RPL [34]. The area under the curve (AUC) of IFN-γ/IL-4 was 0.821 (*p* < 0.05) [34]. Hence, Th1/Th2 ratios may be utilized to predict pregnancy outcomes, closely monitor pregnancy, and swiftly treat patients at risk of miscarriage. Contrarily, a Th1-to-Th2 switch was reported in RPL patients compared to healthy pregnant women of comparable gestational weeks [35], which was different from other studies [36,37,38]. Differences in these studies may be related to the selection of the control group and the timing of the investigation. In the study of Bates et al. [35], 25 pregnant women undergoing voluntary termination of the pregnancy were included but obstetrical and infertility histories of the control group were not reported. Th1/Th2 ratios vary during each developmental phase of pregnancy [39,40], but in the study of Makhseed et al., the study and control groups were investigated in different gestational weeks [38]. It is noteworthy that several animal studies failed to document that a single aberrant cytokine, either Th1 or Th2 cytokine, can affect pregnancy outcome. IL-10 knock-out (KO) mice did not have altered pregnancy outcomes. Additionally, IL-4, IL-5, IL-9, and IL-13 KO mice did not have decreased live birth rate or offspring weight [41]. Therefore, a complicated cytokine network may determine the balance between Th1/Th2 immune responses during pregnancy with a propensity toward Th2 immune response after implantation is over, which in turn, again biased to Th1 immune response at the time of parturition [7].

### 2.1. TNF-α

TNF-α is an important cytokine produced by Th1 cells and is known to have diverse immune-modulatory functions during various stages of human reproduction. TNF-α promotes trophoblast invasion by regulating the expression and secretion of inflammatory factors derived from endometrial stromal cells, such as IL-17 [42] (Figure 1). Moreover, TNF-α induces vascular endothelial growth factor (VEGF) secretion from trophoblasts, which aids in embryo implantation in the uterus (Table 1) [43]. By coordinating angiogenesis, VEGF is crucial for embryo implantation and placental development [43]. In addition, TNF-α stimulates the production of IL-10, which decreases inflammatory activity [44]. After in-vitro fertilization and embryo transfer (IVF-ET), TNF-α producing CD3^+^/CD4^+^ T cell numbers were raised in RPL patients [45]. However, despite having elevated serum TNF-α expression, these individuals had low IL-10 levels [45]. Neuroendocrine factors such as glucocorticoids may affect the serum TNF-α expression but do not produce a secondary IL-10 increase [46]. Therefore, women with RPL may lack neuro-endocrine regulators, resulting in a TNF-α/IL-10 ratio imbalance [45].

Most researchers believe that the serum level of TNF-α in RPL women is greater than in normal pregnant women [37,57,58] (Table 2). In addition, plasma TNF-α levels increase in RPL patients during the first trimester regardless of the outcome of the pregnancy. Its level in secondary RPL was higher than in primary RPL, possibly due to genetic factors [59]. Patients with secondary RPL may have been sensitized by fetal or trophoblast antigen in a prior pregnancy and generated a humoral or cytotoxic reaction to the trophoblast antigen in the subsequent pregnancy, resulting in inflammatory responses in early pregnancy. In contrast, mitogen-activated PBMCs from RPL women with successful pregnancy outcomes exhibited comparable TNF-α levels to those with miscarriages but lower than normal pregnant females [35]. These findings raised concerns about whether TNF-α is detrimental to pregnancy outcomes. TNF-α induces RPL mostly by increasing the expression of pro-apoptotic genes on the embryonic membrane surface [60], which leads to membrane degradation [61]. Additionally, TNF-α may induce antiphospholipid antibody-related placental injury, leading to miscarriage [62].

### 2.2. IFN-γ

IFN-γ is found in the human endometrium [72]. Both inflammatory cells in the decidua and developing embryos release IFN-γ, which maintains the pregnancy by regulating the expression of IL-6, monocyte chemotactic protein (MCP)-1, and macrophage colony-stimulating factor (M-CSF) in early pregnancy [73]. In the rodent model, IFN-γ facilitates implantation through vessel recasting, vascular development at the implantation site, and maintaining decidual tissues during placental development [28]. However, an excessive level of IFN-γ is harmful to the embryo. Most literature has reported significantly increased IFN-γ expression in spontaneous abortion or RPL patients [37,64,74,75], although Bates et al. discovered no difference between RPL women and normal controls [35]. IFN-γ inhibits the production of GM-CSF, which is indispensable for a successful pregnancy [76]. Moreover, IFN-γ influences the expression of proteins involved in the endometrial adhesion of embryos [77].

### 2.3. Transforming Growth Factor (TGF)-β

TGF-β is a multifunctional cytokine with various regulatory roles [78]. It may stimulate the development of Th1 immune cells [79] (Figure 2). At the same time, it controls the cytokine network and maintains maternal immune tolerance [80]. Controversies surround the expression levels of TGF-β in RPL patients. Plasma TGF-β levels in RPL patients have been reported to be significantly higher than in healthy pregnant women [63,81], although contradictory reports were also reported [64,65]. The increased TGF-β1 could inhibit trophoblast invasion by upregulating kisspeptin expression [82].

### 2.4. IL-1β

IL-1β promotes inflammation, induces B cell development and proliferation, and activates NK and T cells [83]. It impacts endometrial growth during embryo implantation [52] and regulates placentation [84]. In addition, IL-1β can promote the expression of matrix metalloproteinase (MMP), thus participating in the regulation of trophoblast invasion [85]. During pregnancy, β-HCG induces the production of IL-1β from decidual T cells [86]. In a retrospective study of 56 RPL patients and 56 normal pregnant women with antinuclear antibody (ANA), IL-1β predicted RPL with a sensitivity of 76.8% and a specificity of 91.1% [87]. This study indicates that IL-1β plays a crucial role in developing RPL and may predict RPL in ANA-positive women [87]. In animal experiments, the pregnancy rate was decreased when IL-1β receptor antagonist was administered prior to embryo transfer [88].

In addition to its primary function in pregnancy, IL-1β contributes to blood clotting. IL-1β increases the amounts of proteins such as ICAM-1, aυβ3, and MCP-1, which increase platelet stability and the risk of thrombosis [89]. The pro-thrombotic condition generated by these proteins causes the production of microthrombi during early pregnancy, ultimately resulting in miscarriage. Therefore, anticoagulant therapy may be useful for RPL patients with elevated IL-1β expression.

### 2.5. IL-33

IL-33 belongs to the IL-1 family [90]. It enhances Th1 and Th2 immune responses [91] and contributes to the Th17 immune response [92]. Additionally, it activates NK cells and promotes IFN-γ production from NK cells [93,94]. During a normal pregnancy, IL-33 enhances Th2 immunity and maintains maternal immune tolerance to the fetus [95].

Serum IL-33 level was considerably lower in RPL women compared to those with normal pregnancies (*p* < 0.05) [66], as validated by another investigation [96]. However, others discovered that the serum IL-33 levels in pregnant women who were going to miscarry were much higher than those of normal pregnancies (>6 weeks of gestation) [97]. The elevated IL-33 may function during pregnancy as a compensating strategy to save the fetus. However, the mechanism by which decreasing IL-33 expression and leading to RPL remains unclear. A successful pregnancy needs the activation of the IL-33/ST2 pathway, and IL-33 is required for embryo implantation in the mice model [56]. Furthermore, IL-33 played a vital role in the placental formation and maintenance of healthy NF-κB and ERK1/2 pathways [98]. These activated pathways upregulate the CCL2/CCR2 axis, subsequently enhancing the proliferation and invasion of decidual stromal cells [98]. By boosting Th2 immunity at the maternal-fetal interface, CCL2 creates a favorable environment for the fetus [98]. The reduced serum IL-33 level in RPL may be attributable to the down-regulated CCL2.

### 2.6. IL-4

IL-4 is a Th2 cytokine, and its level in the culture supernatants of PBMCs from RPL was lower than that of normal pregnant women [35], although a contradictory study was reported [64]. However, the IL-4 level in the peripheral blood of RPL patients was not linked with pregnancy outcomes [35]. 

### 2.7. IL-6

IL-6 is essential for embryo implantation and placentation [99]. During parturition, IL-6 levels were reported to be higher in maternal serum [100], amniotic fluid [101], and placental tissues [102] than without labor. IL-6 has multiple functions, such as promoting inflammatory responses and suppressing TNF-α expression [103]. IL-6 level was significantly higher in the culture supernatant of PBMCs from normal pregnancies than RPL [37]. Additionally, RPL women with low IL-6 levels during early pregnancy had poor pregnancy outcomes [57]. When PBMCs were reconstituted and stimulated with autologous trophoblast or choriocarcinoma cell antigens, leukocytes from RPL patients secreted persistently lower levels of IL-6 than those of healthy controls [36].

### 2.8. IL-10

IL-10 is a Th2 cytokine that suppresses the Th1 immune response by reducing the expressions of TNF-α, IFN-γ, and IL-1 [104]. IL-10 is produced by activated Th2 cells and promotes embryo development by maintaining immunological tolerance [105]. Administration of IL-10 to the abortion-prone mice reduced embryo loss [106], whereas IL-10-deficient mice were more likely to experience miscarriages due to inflammatory changes compared to wild-type mice [107]. Normal pregnant women had higher serum levels of IL-10 compared to RPL patients [57]. The IL-10 level in the peripheral blood of uRPL women was significantly lower, and a higher level was related to healthy pregnancies [36,37,64,67,108]. However, there was a markedly higher amount of IL-10 in the supernatant of cultured PBMCs from pregnant RPL women compared to those of normal pregnant controls, supporting a notion that Th2 immunity was enhanced rather than repressed in RPL [35]. Moreover, the peripheral blood IL-10 level in RPL patients did not predict pregnancy outcomes [35]. IL-10 has a significant role in linking immune responses and angiogenesis and suppresses endoplasmic reticulum (ER) pressure, promoting protein composition and energy stabilization [109]. Increased ER stress leads to the activation of pro-inflammatory responses and placental abnormalities [110].

## 3. Treg and Th17-Related Cytokines in Maternal Peripheral Blood

A Treg/Th17 cell imbalance was identified in RPL women [13]. Compared to healthy pregnant women, RPL patients have fewer Treg cells [111] and more Th17 cells in the peripheral blood and decidua [14,32,69,112,113,114]. Due to their pro-inflammatory activities, Th17-related cytokines, including IL-17, can cause embryo rejection, whereas Treg cell-regulated cytokines such as IL-10 and TGF-β may enhance immunological tolerance and improve pregnancy outcomes [112,115].

### 3.1. IL-6

IL-6 is a crucial cytokine that represses the proliferation of Treg cells and promotes the differentiation of Th17 cells (Figure 3). Serum IL-6 and soluble IL-6 receptor levels were elevated in RPL women, while the inhibitor of IL-6/IL-6R signal transduction, soluble glycoprotein (gp)130, was reduced [116]. Following paternal lymphocyte immunotherapy, the expression of soluble IL-6R was reduced, whereas the soluble gp130 level and Treg cell numbers were increased [116]. These data imply that IL-6 signaling is crucial in regulating the Treg to Th17 cell ratio. Moreover, constant Toll-like receptor (TLR) activation or the suppression of Treg function by IL-6 has been documented [117], indicating that the decline of Treg cell function in RPL patients was associated with the IL-6-related inflammatory response.

### 3.2. IL-7

IL-7 modulates Th17 cell development from naïve T cells [48]. IL-7 was revealed to promote the differentiation of Th17 cells via inhibiting forkhead/winged helix transcription factor (FOXP3) expression in Treg cells and subsequently inhibit the differentiation of Treg cells induced by TGF-β [7,118]. Then, IL-7 disturbs the Treg/Th17 equilibrium in favor of Th17 immunity, ultimately resulting in RPL [119]. Significantly elevated expressions of IL-7 in the decidua of women with spontaneous abortion and RPL indicate that IL-7 likely plays a significant role in promoting pro-inflammatory immune response at the embryo-maternal interface [119]. However, the study was limited by the lack of individual IL-17 investigation, which might be necessary for clinical work [119]. The underlying mechanism of the IL-7 signaling pathway was investigated in RPL by evaluating the IL-7 and IL-7R expression levels in the decidua tissues of RPL patients [32]. By administering an IL-7R inhibitor to the abortion-prone mice, FOXP3^+^ Treg cells (%) were dramatically increased, and Th17 cells (%) were significantly reduced [32]. When IL-7R expression was decreased in endometrial stromal cells, the IL-7 signaling pathway was blocked, and the ability of trophoblast cells to invade was diminished [32]. RPL is partially mediated by the diminished invasion of trophoblast cells [120]. In autoimmune instances, IL-7 expression is increased to compensate for decreased IL-7R [121,122]. Based on research advancements in autoimmune diseases, drugs targeting the IL-7 signaling pathway could be evaluated for the treatment of RPL [119].

### 3.3. IL-17

The IL-17 family consists of multiple related cytokines ranging from IL-17A to IL-17F. Among those, IL-17A and IL-17F are the most widely studied cytokines. Due to its pro-inflammatory function, the possible role of IL-17 in spontaneous abortion has received increasing interest in recent years [69,123,124,125]. However, a recent study revealed that stromal cell-derived IL-17 aggregated Th17 cells, thereby accelerating trophoblast invasion and reducing trophoblast death, indicating a favorable effect on pregnancy maintenance [126]. This result revealed a novel role for Th17 in trophoblast implantation and placental development. Similarly, another study confirmed that the peripheral blood IL-17 levels in normal pregnant women were higher than those of patients with abortion, showing that IL-17 is a protective regulatory factor for a healthy pregnancy [127]. However, the abortion patients in this study were in the first trimester, while the normal pregnant women were in the second or third trimester [127]. In contrast, the morbidity of uRPL is associated with an increased level of IL-17 in peripheral blood [48]. In the mice model, intraperitoneal injection of IL-17 into normal pregnant mice triggered miscarriages, whereas the anti-IL-17 antibody reduced the prevalence of miscarriage in abortion-prone mice [128]. Therefore, the serum level of IL-17 may predict pregnancy outcomes, and further clinical studies are warranted.

### 3.4. IL-22

Previous studies have indicated that IL-22 is advantageous for pregnancy. IL-22 has a role in the healing process of damaged trophoblast cells at the maternal-fetal interface because it participates in epithelial cell regeneration and tissue repair by binding to IL-22R1 on trophoblast cells [129,130]. Significantly fewer IL-22-derived CD4^+^ T cells were demonstrated in the decidua of RPL women who miscarried genetically normal embryos than in normal pregnant women [55]. IL-22 mRNA and its transcription factor are at the implantation site [55]. Therefore, producing IL-22 in the implantation zone is likely essential for a healthy pregnancy. Contrary to these studies, it has been reported that serum IL-22 level was elevated in RPL patients [68]. Further studies are needed to explore the immunopathogenic mechanism of IL-22 in RPL.

### 3.5. IL-23

When the IL-23 level is increased above the initial levels, Th17 cells expand dramatically, resulting in an imbalance of Treg/Th17 cells and consequential embryo rejection [131]. Both peripheral blood and decidual levels of IL-23 were considerably higher in RPL women than in normal pregnancies, indicating that IL-23 may be involved in the pathophysiology of RPL [67,69].

### 3.6. IL-27

Th17 cell expression is inhibited by IL-27, which is present at the maternal-fetal interface in both human and mouse studies [125,132,133,134]. In an experiment using mice, decreased IL-27 expression in decidua was associated with an increased incidence of miscarriage [133]. Decidual IL-27 expression in RPL patients was lower than in women with spontaneous abortion or healthy pregnancies [125]. IL-27 inhibited the level of IL-17 and raised the expression of IL-10 in a dose-dependent manner but had no effect on the expression of TGF-β in decidua [125]. Additionally, IL-27 affects IL-17 and IL-10 levels in women with RPL, which is secreted mostly by Th17 and Treg cells [125]. These findings suggest that IL-27 can influence specific immune cells in the decidua during pregnancy.

### 3.7. IL-35

CD4^+^ Foxp3^+^ Treg cells produce the majority of IL-35 [135], and trophoblast cells also secrete IL-35 during early pregnancy [136]. IL-35 is critical for maintaining the inhibitory action of Treg cells [135]. The expression of IL-35 in the serum of RPL patients was considerably lower than that of normal pregnant women, indicating that IL-35 has a role in maintaining a pregnancy [70,71].

According to the studies mentioned above, the expressions of Th1 cytokines in the peripheral blood of RPL patients are significantly increased, including TNF-α and IFN-γ. On the other hand, the circulating levels of Th2 cytokines, such as IL-6 and IL-10, were remarkably lower in RPL females compared to normal pregnant women. IL-10 seems to be a core Th2 cytokine since it inhibits the activity of Th1 cytokines and has the function of immune regulation. Nowadays, maternal-fetal immune tolerance has gradually expanded from the Th1/Th2 pattern to Th1/Th2/Treg/Th17 pattern. Therefore, Th17 cytokines IL-17 and IL-22 may play central roles in the etiology of RPL, which needs further research.

## 4. Endometrial Cytokine Imbalance in RPL

The imbalance of cytokine expression in the endometrium or decidua is relevant for RPL (Table 3). In both women with RPL and abortion-prone animal models, an inordinate expression of a few cytokines associated with implantation and early embryo development was present in immune and non-immune cells in the endometrium or decidua. This creates an adverse cytokine milieu, which assaults immunological tolerance and causes fetal rejection. Due to the intricacy of cytokines in the endometrium, which undergo dramatic changes during pregnancy, it is difficult to define the precise function of each cytokine in RPL. In addition, it is hard to identify the source of a specific cytokine because numerous immune and non-immune cells can produce the same cytokine. Despite the limitations highlighted, some cytokines have been reported to play a significant role in this illness.

Before the blastocyst reaches the endometrium, stromal cells produce TNF-α and IL-1β to trigger an inflammatory response during the implantation window [49]. Human endometrial cells can express TNF-α mRNA [142,143]. The expression of TNF-α varies throughout the menstrual cycle and gradually increases in the late luteal phase [144]. In addition, inhibiting TNF-α reduces stress-induced miscarriage in mice [145]. RPL patients had elevated levels of IFN-γ mRNA and protein in their endometrium and decidua [138]. TNF-α and IFN-γ can impede embryo growth by inducing apoptosis of trophoblast cells [146].

In the first trimester, IL-1 family cytokines are predominant at the maternal-fetal interface [76]. IL-1β contributes to decidualization by increasing the production of cyclooxygenase-2 (COX2) and prostaglandin E2 (PGE2) [85]. In a study of placental tissues (6–13 weeks of gestation) taken from 15 normal pregnancies and 15 RPL patients, significantly elevated levels of IL-1β were demonstrated in the decidua of the RPL group [31]. These results illuminated the complicated function of IL-1β at the maternal-fetal interface and its effect on abortion [31]. Furthermore, the protein expression of IL-18 in decidual tissues of RPL patients was much higher than that of normal pregnant women, indicating that precise regulation of IL-18 was necessary for a healthy pregnancy [141].

IL-6 plays a constructive role in decidualization and increases endometrial receptivity, thus facilitating the regulation of maternal-fetal interaction [85]. Patients with RPL have lower IL-6 mRNA and protein expressions in the mid-luteal endometrial tissues [139,147,148]. However, another study showed that IL-6 mRNA and protein expressions in the decidual tissues of RPL patients were elevated compared to normal pregnant controls [137]. The disparities in these studies may be attributable to the heterogeneity of the different populations and the various underlying disorders.

IL-10 expression at the maternal-fetal interface was documented [149,150]. It is a potent cytokine that induces the production of tolerogenic DCs [151], which are essential for sustaining maternal-fetal immunological tolerance [152]. Patients with RPL whose decidual tissues express low levels of IL-10 have an impaired immune protection system [138,153,154,155]. Decreased production of IL-10 with the increased synthesis of inflammatory factors, may provide a condition that induces early miscarriage [156]. In addition, IL-10 and other cytokines, such as TGF-β, can help coordinate the development of Treg cells in the decidua [157]. Furthermore, the expression of IL-4 was remarkably lower in the decidua of RPL patients compared with normal pregnancies [138,158]. However, the IL-17A expression in the decidua tissue of RPL patients was considerably higher than in healthy pregnant women [112].

A recently published report indicates that IL-35 is expressed in primary early trophoblast cells and the HTR-8 trophoblast cell line. Trophoblasts could transform naive conventional T cells into iTR 35 cells in the presence of IL-35. Murine spontaneous abortion models have decreased expressions of both IL-35 and iT_R_ 35 cells at the fetomaternal interface, suggesting that IL-35 is crucial for sustaining fetomaternal tolerance [159].

IL-11 is a crucial cytokine associated with decidualization [160] and placentation [161]. The reduced expression of IL-11 in the endometrial tissues of RPL shows that it plays a significant role in preventing abortion [140]. Therefore, RPL might be potentially addressed by correcting IL-11 signaling deficiencies in the endometrium [162]. Defective decidualization and reduced proliferation of uterine stromal cells were observed in IL-11R KO animals [163,164]. However, these in vivo intervention experiments conducted in mice can not fully reflect changes in humans. Defective implantation and placentation are major concerns in human pregnancy since spontaneous miscarriage occurs in 30% of human pregnancies, most happening before a clinical pregnancy test [165]. In addition, IL-11 is expressed in the human endometrium [166]. Thus, it is necessary to conduct a relevant investigation to identify whether the changes in IL-11 and its receptors are related to RPL.

It is known that RPL is related to a reduction in decidual TGF-β compared to healthy pregnant women due to a decreased production of TGF-β from decidual DCs [153], a decreased mRNA expression of TGF-β [137], or a smaller proportion of TGF-β^+^ Tregs in the decidua [154]. Furthermore, when trophoblast cells obtained after uterine curettage were investigated, those from RPL women (*n* = 11) had substantially decreased expression of TGF-β3 compared to those from normal pregnant women (*n* = 20) [167]. Since TGF-β is involved in stimulating implantation and early embryo development, a reduction in TGF-β may be causally related to the immunological etiology of RPL. Furthermore, a recent study showed that the TGF-β signaling pathway promotes the differentiation of placental EVTs into decidual EVTs, playing a vital role in decidualization [168]. 

According to a previous report [169], Th1 cytokines stimulate pro-coagulant factors on vascular endothelial cells, leading to the production of thrombus and inflammation at the maternal-fetal interface. When fibrinogen-like protein 2 (fgl2), one of the pro-coagulants, is blocked by anti-fgl2, spontaneous abortions were prevented in the mice model. In human pregnancies, enhanced fgl2 expression was demonstrated in the trophoblast cells of failed human pregnancies [170]. Fgl2 plays a role in the conversion of prothrombin to thrombin, the accumulation of fibrin, and the activation of polymorphonuclear leukocytes, which in turn impairs placental blood flow. Th2 cytokines suppress and counteract the Th1 response. In addition, Th1 cytokines can promote apoptosis and destroy the trophoblast barrier between a semi-autogenous fetus and the maternal immune system, leading to miscarriage. Th1 cytokines can also exert their effects by increasing the proliferation of NK cells, lymphokine-activated killer (LAK) cells, and cytotoxic T lymphocyte (CTL) cells, which destroy trophoblast cells and result in fetal absorption in murine pregnancies [171].

## 5. Cytokines Produced by NK Cells

Similar to T cells, NK cells can be categorized based on the cytokines they secrete [172,173,174,175]. In the peripheral blood of non-pregnant women, IFN-γ or TNF-α-producing and IL-4, IL-5, IL-13, TGF-β or IL-10-non-producing CD56^bright^ NK cells and CD56^dim^ NK cells (NK1) are the most abundant cells (60%) [176]. However, the proportion of IL-4, IL-5, and IL-13-producing, IFN-γ, TNF-α, IL-10, or TGF-β-non-producing NK2 cell populations is extremely low [176]. NK3 cells refer to the TGF-β-producing, IFN-γ, TNF-α, IL-4, IL-5 or IL-13-non-producing cells, and the IL-10-producing cells might be NKr1 cells [176]. In a normal pregnancy, NK cells switch from type 1 to type 2 immune response [177,178]. The numbers of circulating NKr1 and decidual NK3 cells are reduced in spontaneous abortion patients, suggesting that these cells are key players in maintaining pregnancy by regulating maternal immune response [176].

Elevated NK cell cytotoxicity has been reported in the peripheral blood of RPL patients [179]. In RPL patients, NK cytotoxicity may be utilized as a biomarker to predict future abortions [180,181]. However, a contradictory study was reported [182], which could be attributed to the different experimental methodologies and study designs.

RPL patients had a considerably greater number or proportion of circulating NK cells than normal pregnant women [183,184,185]. In women with RPL, endometrial NK cell counts were significantly higher during the mid-luteal phase compared to the normal fertile controls [186]. dNK cells exhibited increased NK1/NK2 ratios (IFN-γ/IL-4 producing NK cell ratios) in women with RPL [187]. However, TNF-α and IFN-γ producing uNK cells in RPL patients were lower than in normal pregnant women [188]. The proportion of IL-4 and IL-10-producing CD56^bright^ NK cells in the peripheral blood was lower in non-pregnant women with a history of RPL than in normal fertile women [189]. In contrast, the proportion of TNF-α and IFN-γ producing NK cells increased significantly [189]. These findings indicate the presence of increased NK1/NK2 ratios in women with RPL [189].

Moreover, RPL patients have higher absolute numbers of CD56^+^ NK cells with intracellular IFN-γ, TGF-β, and IL-4 expressions, demonstrating that the immune system is activated in RPL women compared to normal controls [190]. Decidual NK22 cells are the primary cells to produce IL-22. IL-22 participates in host defense, mucosal homeostasis, and trophoblast cell invasion [191]. CD56^bright^/IL-22^+^ cells were negatively correlated with CD56^bright^ NK cells producing IFN-γ or TNF-α both in peripheral blood and endometrium [191]. The mRNA and protein expressions of IL-22 in the decidua of RPL were considerably reduced [192], and diminished IL-22 levels in the decidual tissues of RPL may compromise decidual homeostasis and ultimately result in embryo rejection [192].

### 5.1. Cytokine in the Micro Milieu of NK Cells

The maternal-fetal interface contains IL-15 and IL-18, activating NK cells to secrete various cytokines. IL-15 is also present in the female reproductive tract [193] and stimulates the proliferation of uNK cells in both mice and humans [194,195]. By activating progesterone receptors, stromal cells released IL-15, and the level of IL-15 was positively correlated with the number of uNK cells [196]. In animal studies, uNK cells cannot differentiate in the uterus without IL-15 [197]. Similarly, IL-15 KO mice lacked uNK cells. However, a lack of uNK cells did not impact on embryo implantation and development [197]. Endometrial stromal cells produce more IL-15 during decidualization [198]. In humans, higher endometrial IL-15 expression was closely associated with RPL [199]. Similarly, IL-15 mRNA and protein expressions were significantly increased in the placental tissues of RPL patients [200]. Over-expression of IL-15 in the decidua of RPL patients may lead to the failure of implantation and angiogenesis, followed by placental injury and fetal loss [200].

IL-18 can stimulate the production of IFN-γ from uNK cells with the presence of Th1 cytokine IL-12 [201]. It may augment the level of pro-inflammatory cytokines released by macrophages at the embryo-maternal interface, followed by the activation of uNK cells. Additionally, IL-18 increases NK cell cytotoxicity [202] and stimulates Th2 cytokine production from NK cells, such as IL-4 and IL-13 [203]. IL-18 expressions in peripheral blood and endometrium of RPL patients were considerably greater than those of healthy fertile females [204]. Briefly, cytokine variations in the micro milieu of NK cells may influence NK cell activity and contribute to the pathogenesis of RPL.

### 5.2. uNK Cells and Angiogenic Cytokines

RPL is associated with increased endometrial vascular density, suggesting that RPL is implicated in the premature maturation of the vascular network [205]. It has been reported that uNK cells produce extensive angiogenic factors, such as VEGF-C, angiopoietin (Ang), and platelet-derived growth factor (PLGF), which are vital for angiogenesis in the endometrium [29]. During the implantation window, CD56^+^ uNK cells are one of the major sources of angiogenic cytokines in the endometrium [206]. Angiogenic cytokine gene array study demonstrated that the expressions of VEGF-A, Ang, and basic fibroblast growth factor (bFGF) in CD56^+^ uNK cells from RPL patients were considerably higher than those of normal pregnant women [206]. Differences in results may be attributed to the test method (angiogenic cytokine array vs. ELISA), experiment sensitivity, and study participants.

## 6. Cytokine Gene Polymorphisms in RPL

The increased expressions of pro-inflammatory cytokines may be caused partly by gene polymorphisms, and women with cytokine gene polymorphisms might be genetically prone to RPL. In addition, gene polymorphisms may affect the expression levels of cytokines, as well as their functions, which play an important role in the pathogenesis of RPL [207]. Therefore, cytokine gene polymorphisms might be utilized to predict abortion, stratify pregnancy management, and improve maternal and fetal pregnancy outcomes by increasing the surveillance of the high-risk abortion group.

### 6.1. TNF-α Gene Polymorphisms in RPL

In RPL patients, polymorphisms in the TNF-α gene have been extensively investigated (Table 4). Previous meta-analyses indicated that TNF-α -308G/A and -238G/A polymorphisms had no relationship with the risk of RPL [208,209]. However, subsequent meta-analyses demonstrated that the TNF-α -308G/A polymorphism was related to an increased risk of RPL [210,211] but not for TNF-α -238G/A [210,211].

### 6.2. IFN-γ Gene Polymorphisms in RPL

IFN-γ 874A > T gene polymorphism affects the level of IFN-γ, and the T allele increases the cytokine expression [223]. Genotype frequency of IFN-γ 874A > T polymorphism was significantly different in Argentine women with RPL from that of healthy pregnant women [221]. The frequency of AT genotype was significantly higher, and that of the AA genotype was considerably lower in patients with RPL compared with those of normal fertile women [221]. It has been observed that the TT genotype that increases IFN-γ production conspicuously boosts the risk of RPL [219]. However, other studies reported a lack of association between IFN-γ 874A > T polymorphism and RPL [207,208,222].

### 6.3. IL-10 Gene Polymorphisms in RPL

Different alleles at sites −1082, −592, and −819 regulate IL-10 expression [224,225]. The IL-10 −1082 GG, −592 AA, and −819 TT genotypes may enhance IL-10 expression [224,225]. It is hypothesized that phenotypes related to the underproduction of IL-10 may induce pregnancy complications. Certain IL-10 genotypes increase the chance of RPL, whereas others protect against abortion (Table 5). A recent meta-analysis of 2047 RPL patients and 2055 control pregnant women revealed an association between −1082 A/G and the incidence of RPL [226]. However, no significant association was seen between the −819C/T, 592C/A genotypes and RPL [208,226]. Four meta-analyses produced contradictory results about the association between the most extensively investigated −1082 A/G genotype and RPL. The first meta-analysis comprised six studies with 635 RPL patients and concluded that −1082 A/G1082 was unrelated to RPL [208]. The second meta-analysis comprised six studies and reached the same conclusion [227]. The third meta-analysis, which included the previous six studies [208,227] and their own data from India, indicated that the wild-type allele of −1082 A/G was protective for RPL [228]. A recent publication of the fourth meta-analysis, including 13 studies, suggested that the GG genotype increased the risk of RPL [229]. The conflicting results may be related to the diverse study populations.

### 6.4. Other Cytokines Gene Polymorphisms in RPL

Gene polymorphisms in other cytokines are summarized in Table 6 (IL-18), Table 7 (IL-6), and Table 8 (IL-1β and IL-1 RN). Furthermore, no significant correlation was present between the two IL-4 polymorphisms and the probability of RPL [222,233]. Additionally, women who carry the IL-2RN*2 allele are less likely to have clinical pregnancies after IVF [234].

Inconsistent results in expression levels and gene polymorphisms of various cytokines in women with RPL could be due to other factors affecting the expression levels of cytokines, differences in study design, study population, and lifestyle. Moreover, it is vital to investigate whether the reported genotype frequency of particular cytokine polymorphism can be applicable to RPL of other ethnic backgrounds, as some of the studies mentioned above only applied to a certain ethnic population and not others [250]. Multiple cytokine polymorphisms may contribute to RPL rather than a single gene polymorphism. Moreover, altered cytokine production by gene polymorphisms may not represent the immune and inflammatory change at the maternal-fetal interface since cytokines exert their effect in both autocrine and paracrine manners. Finally, the association between cytokine gene polymorphisms and RPL warrants further studies with larger study populations. Meta-analysis may solve the decreased statistical significance caused by the small sample size and obtain more accurate results.

## 7. Hormonal Regulation of Cytokines and RPL

Various hormones regulate the expression of pregnancy-maintaining Th2 cytokines. The decrease of estrogen (E2) and progesterone (P4) induced the production of Th1 pro-inflammatory cytokines [251]. In addition, E2 and P4 enhance Th2 anti-inflammatory response and reduce the activity of pro-inflammatory macrophages and NK cells [252]. P4 levels are similar at the maternal-fetal interface throughout implantation and early pregnancy. In humans, P4 promotes the cytokines production by T cells, such as IL-4, IL-5, leukemia inhibitory factor, and M-CSF in decidua [253], while suppressing the production of IFN-γ and TNF-α [254,255] and impeding the production of Th17 related cytokines, such as IL-17A and IL-23 in peripheral blood [256]. A few studies investigated the indications for the use of P4 in treating RPL. A large sample study involving 826 subjects revealed that compared with the placebo group, vaginal micronized progesterone plays a weak or no difference in improving the living birth rate of RPL patients [257]. Similarly, the proof of dydrogesterone for treating RPL is low certainty compared to placebo [257]. There is no sufficient data to analyze the effect of 17-α-hydroxyprogesterone or oral micronized progesterone on increasing the living birth rate of RPL patients [257]. In another study, the subgroup analysis of three experiments contained patients with three or more recurrent miscarriages [258]. P4 significantly reduced the abortion rate compared with placebo or no therapy [258]. It seems that P4 has a significant therapeutic effect on RPL. Therefore, they pointed out that P4 could be considered for treating RPL women since the abortion rate was lower and there was no augment in side effects of mothers and neonates [258].

E2 also has immunomodulatory effects. E2 levels during pregnancy have been shown to affect the Th1/Th2 cytokine balance, maintaining maternal-fetal tolerance [259]. E2 regulates cytokine expression in a dose-dependent manner. It increases IFN-γ and IL-12 levels in the peripheral blood before ovulation. However, it increases IL-10 levels throughout pregnancy, reducing the IFN-γ/IL-10 ratio during human pregnancy [260]. In the human PBMC culture study, E2 decreased the secretion of IL-2 and IFN-γ but did not affect the level of TNF-α, IL-17A, and IL-23 [256]. However, E2 administration significantly reduced the circulating IL-17 expression in allergic encephalitis mice [261]. These discrepant results imply that it is necessary to explore the effect of E2 on Th17 cytokines in pregnant women. Like P4, E2 increases peripheral blood IL-6 levels in humans [256]. Since E2 can suppress the serum levels of Th1 cytokines, such as IL-2 and IFN-γ, and increase the serum level of Th2 cytokine IL-6, it can regulate the cytokine balance in favor of the Th2-type immune response.

## 8. Conclusions

RPL is a disorder with numerous potential causes. Dysregulation of cytokine expression may be one of the underlying pathologies, but the mechanisms need to be explored further. Cytokines rarely exhibit activity independently; rather, they comprise a regulatory network that maintains equilibrium. Moreover, the ultimate effect of an altered cytokine level is determined by the overall modulation of the immune response rather than a mechanism by a specific cytokine. The gaps between the highest and lowest levels of a particular cytokine can be as much as tenfold; hence, one mechanism by a single cytokine cannot account for RPL. Consequently, understanding the intricate mechanisms of various cytokines may aid in establishing therapeutic modalities for RPL of immune etiologies and achieving a successful pregnancy.

Currently, available data do not support a routine check for cytokine or cytokine gene polymorphisms in women with RPL, and most obstetrical guidelines do not recommend testing cytokine levels [250]. It may be unnecessary to do cytokine testing in RPL patients who clearly have an endocrinologic or infectious etiology. Noteworthily, Th1/Th2 cell ratios have been utilized to diagnose and monitor women with RPL. Even though a variety of immunomodulatory treatments, including corticosteroids, paternal lymphocyte immunization, progesterone, intralipid infusion, TNF-α inhibitors, granulocyte colony-stimulating factor (G-CSF), low molecular weight heparin (LMWH), and intravenous immunoglobulin (IVIg) therapy, have been utilized in the treatment of RPL women, their efficacy is still the subject of contention [262,263,264]. Hence, not a single gene or a cytokine but a panel of cytokines and cytokine genes is likely to be necessary to assess women with RPL, considering the complexity of the immune network and the multifactorial nature of treatment responses. In women with RPL and dysregulated cytokine profiles, local or systemic administration of cytokines or their antagonists may be a potential therapeutic approach for maintaining immunological balance and improving maternal and fetal outcomes. Well-designed large-scale clinical trials are needed in the future.

## Figures and Tables

**Figure 1 ijms-24-00132-f001:**
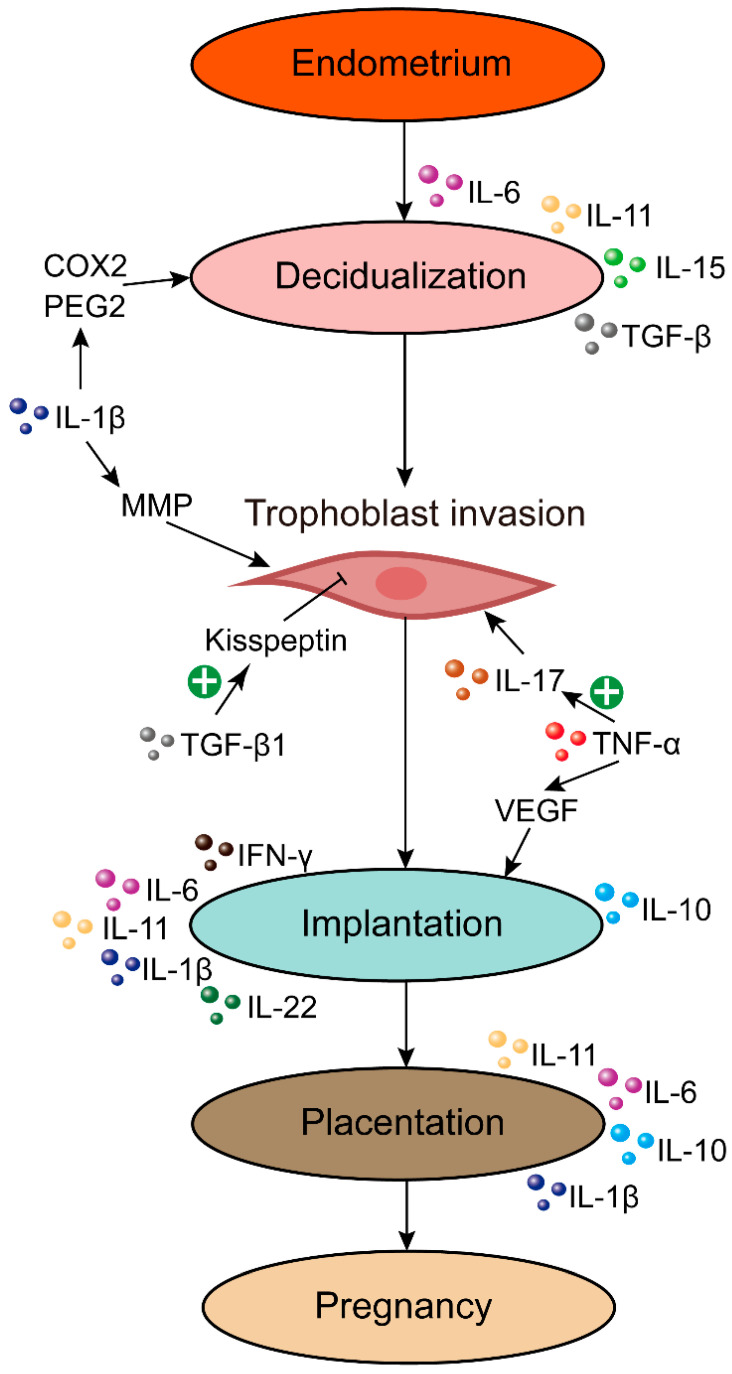
The roles of pro- and anti-inflammatory cytokines in early pregnancy. IL-6, IL-11, IL-15, and TGF-β are involved in regulating decidualization. Furthermore, IL-1β contributes to decidualization by elevating the generation of COX2 and PGE2. Furthermore, IL-1β elevates the expression of MMP, a critical regulator of trophoblast invasion. TGF-β1 reduces the invasion of trophoblast cells by enhancing kisspeptin expression. TNF-α induces trophoblast invasion through increasing IL-17 production from endometrial stromal cells. Moreover, TNF-α plays an important role in implantation by inducing VEGF generation from trophoblast cells. IFN-γ, IL-6, IL-10, IL-11, IL-1β, and IL-22 are associated with embryo implantation. In addition, IL-6, IL-10, IL-11, and IL-1β participate in placentation.

**Figure 2 ijms-24-00132-f002:**
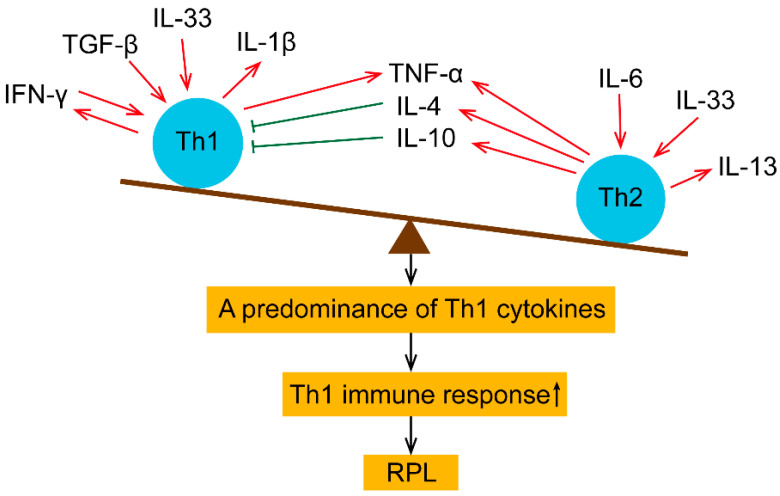
Schematic diagram of the unbalanced Th1/Th2 cytokine network in peripheral blood of RPL patients. TNF-α can be produced by Th1 or Th2 cells. IFN-γ is secreted by Th1 cells, which induce the differentiation of Th1 cells. Moreover, TGF-β augments the differentiation of Th1 cells, which release IL-1β. In addition, IL-33 can enhance Th1 or Th2 differentiation. Th2 cells produce IL-4 and IL-13. IL-4 is known to compromise the production of Th1 cells. IL-6 is found to promote Th2 cells. IL-10 is a Th2 cytokine that inhibits the Th1 immune response.

**Figure 3 ijms-24-00132-f003:**
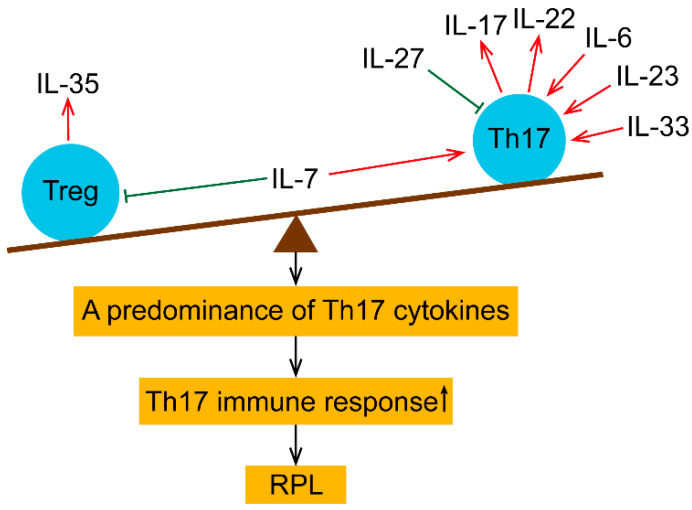
An illustration of the imbalanced Treg/Th17 cytokines in the periphery of RPL patients. IL-35 is predominantly produced by Treg cells. IL-7 promotes the differentiation of Th17 cells and compromises Treg cells, promoting inflammatory responses. Th17 cells secrete IL-17 and IL-22. IL-6 induces the differentiation of Th17 cells. In addition, IL-23 increases the number of Th17 cells. IL-33 contributes to the Th17 immune response. On the contrary, IL-27 impedes the expression of Th17 cells. Accordingly, aberrant expression of these cytokines in peripheral blood may cause Treg/Th17 immune imbalance, triggering the pro-inflammatory immune response and eventually leading to RPL.

**Table 1 ijms-24-00132-t001:** Cytokines exert anti- or pro-implantation properties.

Anti-Implantation	
IL-6	[47]
IL-17	[48]
IL-23	[48]
**Pro-Implantation**	
TNF-α	[49]
IFN-γ	[28]
TGF-β1	[50]
IL-1	[51]
IL-1β	[52]
IL-10	[53]
IL-11	[54]
IL-22	[55]
IL-33	[56]

**Table 2 ijms-24-00132-t002:** Alternations of cytokines in peripheral blood of RPL women compared with normal females.

Cytokines	Expression	Number of the Experimental Group	Number of the Control Group	Reference
TNF-α	Increased	23	24	[37]
IFN-γ	Increased	23	24	[37]
TGF-β	Increased	36	30	[63]
	No difference	32	32	[64]
	No difference	29	27	[65]
IL-33	Decreased	142	123	[66]
IL-4	Increased	19	15	[35]
	No difference	32	32	[64]
IL-6	Decreased	23	24	[37]
IL-10	Decreased	23	28	[57]
	Decreased	19	16	[36]
	Decreased	23	24	[37]
	Decreased	32	32	[64]
	Decreased	30	30	[67]
IL-17	Increased	20	20	[48]
IL-22	Increased	46	28	[68]
IL-23	Increased	30	30	[67]
	Increased	15	15	[69]
IL-35	Decreased	60	40	[70]
	Decreased	40	120	[71]

**Table 3 ijms-24-00132-t003:** Changes of cytokines in endometrium/decidua of RPL patients compared with healthy women.

Cytokines	Expression	Number of the Experimental Group	Number of the Control Group	Reference
TNF-α	Increased	35	40	[137]
IFN-γ	Increased	10	10	[138]
TGF-β	Decreased	35	40	[137]
IL-1β	Increased	15	15	[31]
IL-4	Decreased	10	10	[138]
IL-6	Increased	35	40	[137]
	Decreased	9	12	[139]
IL-10	Decreased	10	10	[138]
IL-11	Decreased	16	9	[140]
IL-17	Increased	15	15	[69]
IL-18	Increased	15	9	[141]
IL-23	Increased	15	15	[69]

**Table 4 ijms-24-00132-t004:** Studies for TNF-α gene polymorphisms and RPL.

Gene Polymorphism	Authors	Race	Number of Study and Control Groups	Genotype		Test Method	Conclusions
				Study Group	Control Group		
−238G/A	Zammiti, W. et al., 2009 [212]	Tunisian	372 vs. 274	264/88/20 ^a^	215/52/7 ^a^	PCR-RFLP	The polymorphism of the −238G/A gene was associated with the occurrence of RPL.
	Finan, R. et al., 2010 [213]	Bahraini Arabs	204 vs. 248	148/52/4 ^a^	200/48/0 ^a^	PCR-RFLP	−238G/A variants were independent risk factors for RPL.
	Liu, C. et al., 2010 [214]	Chinese	132 vs. 152	128/4/0 ^a^	135/17/0 ^a^	PCR	A statistical difference was exhibited in −238G/A polymorphism.
	Gupta, R. et al., 2012 [215]	Indian	300 vs. 500	121/63/16 ^a^	154/113/33 ^a^	PCR-RFLP	RPL women tended to carry the G allele.
	Alkhuriji, A. et al., 2013 [216]	Saudis	65 vs. 65	57/8/0 ^a^	55/7/3 ^a^	PCR	NS
	Lee, B. et al., 2013 [217]	Korean	357 vs. 236	330/26/1 ^a^	228/8/0 ^a^	PCR-RFLP	TNF-α −238G > A variants elevated the incidence of RPL.
	Ma, J. et al., 2017 [207]	Chinese	775 vs. 805	732/41/2 ^a^	745/57/3 ^a^	PCR-RFLP	NS
−308G/A	Babbage, S. et al., 2001 [218]	Caucasian	43 vs. 73	30/13 ^b^	56/17 ^b^	PCR	NS
	Daher, S. et al., 2003 [219]	Brazilian	48 vs. 108	36/12 ^b^	89/19 ^b^	PCR	NS
	Pietrowski, D. et al., 2004 [220]	Caucasian	168 vs. 212	133/33/2 ^a^	167/41/4 ^a^	PCR	NS
	Prigoshin, N. et al., 2004 [221]	Argentinean	41 vs. 54	35/6 ^b^	49/5 ^b^	PCR-SSP	NS
	Kamali-Sarvestani, E. et al., 2005 [222]	Iranian	139 vs. 143	117/14 ^b^	122/21 ^b^	PCR	NS
	Zammiti, W. et al., 2009 [212]	Tunisian	372 vs. 274	319/39/14 ^a^	222/47/5 ^a^	PCR-RFLP	NS
	Finan, W. et al., 2010 [213]	Bahraini Arabs	204 vs. 248	164/32/8 ^a^	212/32/4 ^a^	PCR-RFLP	NS
	Liu, C. et al., 2010 [214]	Chinese	132 vs. 152	110/22/0 ^a^	138/13/1 ^a^	PCR	NS
	Gupta, R. et al., 2012 [215]	Indian	300 vs. 500	229/62/9 ^a^	425/70/5 ^a^	PCR-RFLP	The A allele was more likely to be present in RPL women.
	Alkhuriji, A. et al., 2013 [216]	Saudis	65 vs. 65	33/24/8 ^a^	47/14/4 ^a^	PCR	The −308G > A gene polymorphisms were associated with RPL.
	Lee, B. et al., 2013 [217]	Korean	357 vs. 236	319/36/2 ^a^	213/21/2 ^a^	PCR- RFLP	NS
	Ma, J. et al., 2017 [207]	Chinese	775 vs. 805	683/86/6 ^a^	726/76/3 ^a^	PCR- RFLP	NS
−1031T/C	Finan, R. et al., 2010 [213]	Bahraini Arabs	204 vs. 248	152/36/16 ^a^	219/29/0 ^a^	PCR-RFLP	The frequency of the −1031C allele in RPL was significantly increased.
	Lee, B. et al., 2013 [217]	Korean	357 vs. 236	230/115/12 ^a^	191/45/0 ^a^	PCR-RFLP	TNF-α − 1031T > C variants augmented the chance of experiencing RPL.
−376G/A	Finan, R. et al., 2010 [213]	Bahraini Arabs	204 vs. 248	174/20/10 ^a^	226/22/0 ^a^	PCR-RFLP	Patients with RPL had a higher incidence of carrying the −376A allele.

^a^ Genotype for TNF-α − 238G/A, GG/GA/AA; for TNF-α − 308G/A, GG/GA/AA; for TNF-α − 1031T/C, TT/TC/CC; for TNF-α − 376G/A, GG/GA/AA; ^b^ Genotype for TNF-α − 308G/A, GG/GA+AA; PCR, polymerase chain reaction; RFLP, restriction fragment length polymorphism; SSP, sequence specific primers; NS, no significance.

**Table 5 ijms-24-00132-t005:** Studies for IL-10 gene polymorphisms and RPL.

Gene Polymorphism	Authors	Race	Number of Study and Control Groups	Genotype		Test Method	Conclusions
				Study Group	Control Group		
−1082A/G	Babbage, S. et al., 2001 [218]	Caucasian	43 vs. 73	8/23/12 ^a^	20/41/12 ^a^	PCR	NS
	Karhukorpi, J. et al., 2001 [230]	Finnish	38 vs. 131	13/16/9 ^a^	44/64/23 ^a^	PCR	NS
	Daher, S. et al., 2003 [219]	Brazilian	43 vs. 104	13/19/11 ^a^	45/43/16 ^a^	PCR	NS
	Kamali-Sarvestani, E. et al., 2005 [222]	Iranian	139 vs. 143	62/41/24 ^a^	62/47/21 ^a^	PCR-RFLP	NS
	Zammiti, W. et al., 2006 [231]	Tunisian	350 vs. 200	87/185/72 ^a^	54/107/39 ^a^	PCR-ASA	NS
	Parveen, F. et al., 2013 [228]	Indian	200 vs. 300	86/99/15 ^a^	180/108/12 ^a^	PCR	The A allele has a significant protective effect.
	Kim, J. et al., 2014 [232]	Korean	385 vs. 232	333/50/2 ^a^	198/34/0 ^a^	PCR-RFLP	NS
	Ma, J. et al., 2017 [207]	Chinese	775 vs. 805	683/88/4 ^a^	685/113/7 ^a^	PCR-RFLP	NS
−819C/T	Kamali-Sarvestani, E. et al., 2005 [222]	Iranian	139 vs. 143	77/49/13 ^a^	61/56/15 ^a^	PCR-RFLP	NS
	Zammiti, W. et al., 2006 [231]	Tunisian	350 vs. 200	182/120/48 ^a^	124/57/19 ^a^	PCR-ASA	The −819C/T polymorphisms were genetically associated with RPL.
	Parveen, F. et al., 2013 [228]	Indian	200 vs. 300	59/111/30 ^a^	122/142/36 ^a^	PCR	RPL females tended to own the CT genotype.
−592C/A	Kamali-Sarvestani, E. et al., 2005 [222]	Iranian	139 vs. 143	83/35/14 ^a^	61/56/15 ^a^	PCR-RFLP	−592C/A polymorphism variants were one of the genetic causes of RPL.
	Zammiti, W. et al., 2006 [231]	Tunisian	350 vs. 200	206/93/51 ^a^	134/41/25 ^a^	PCR-ASA	NS
	Parveen, F. et al., 2013 [228]	Indian	200 vs. 300	91/79/30 ^a^	148/116/36 ^a^	PCR	NS

^a^ Genotype for IL-10 − 1082A/G, AA/AG/GG; for IL-10 − 819C/T, CC/CT/TT; for IL-10 − 592C/A, CC/CA/AA; ASA, allele-specific amplification.

**Table 6 ijms-24-00132-t006:** Researches about IL-18 gene polymorphisms in RPL.

Gene Polymorphism	Authors	Race	Number of Study and Control Groups	Genotype		Test Method	Conclusions
				Study Group	Control Group		
−607C/A	Naeimi, S. et al., 2006 [235]	Iranian	102 vs. 103	37/23/42 ^a^	32/17/54 ^a^	PCR	NS
	Ostojic, S. et al., 2007 [236]	Slovenian	125 vs. 136	43/68/14 ^a^	41/79/16 ^a^	PCR	NS
	Yue, J. et al., 2015 [237]	Chinese	484 vs. 468	87/216/181 ^a^	79/211/178 ^a^	PCR	NS
−137G/C	Naeimi, S. et al., 2006 [235]	Iranian	102 vs. 103	57/40/5 ^a^	56/39/8 ^a^	PCR	NS
	Ostojic, S. et al., 2007 [236]	Slovenian	125 vs. 136	59/54/12 ^a^	62/63/11 ^a^	PCR	NS
	Al-Khateeb, G. et al., 2011 [238]	Bahraini	282 vs. 283	146/98/38 ^a^	152/113/24 ^a^	PCR	NS
	Messaoudi, S. et al., 2012 [239]	Tunisian	235 vs. 235	122/82/31 ^a^	126/92/19 ^a^	PCR	NS
	Yue, J. et al., 2015 [237]	Chinese	484 vs. 468	338/108/38 ^a^	357/102/9 ^a^	PCR	−137G/C variants had statistical relationship with RPL in additive and recessive genetic models.
−656C/A	Al-Khateeb, G. et al., 2011 [238]	Bahraini	282 vs. 283	80/144/58 ^a^	140/119/30 ^a^	PCR	−656C/A variants were related to RPL.
	Messaoudi, S. et al., 2012 [239]	Tunisian	235 vs. 235	66/120/49 ^a^	114/97/24 ^a^	PCR	The genotype frequency of −656C/A was significantly correlated with the occurrence of RPL.
−119A/C	Al-Khateeb, G. et al., 2011 [238]	Bahraini	282 vs. 283	157/99/26 ^a^	155/109/25 ^a^	PCR	NS
	Messaoudi, S. et al., 2012 [239]	Tunisian	235 vs. 235	132/82/21 ^a^	127/89/19 ^a^	PCR	NS
−105G/A	Al-Khateeb, G. et al., 2011 [238]	Bahraini	282 vs. 283	98/94/90 ^a^	146/110/33 ^a^	PCR	−105G/A variant was prominently associated with RPL.
	Messaoudi, S. et al., 2012 [239]	Tunisia	235 vs. 235	82/78/75 ^a^	120/89/26 ^a^	PCR	The genotype frequency of −105G/A was significantly different between the two groups.
	Yue, J. et al., 2015 [237]	Chinese	484 vs. 468	349/129/6 ^a^	332/128/8 ^a^	PCR	NS

^a^ Genotype for IL-18 − 607C/A, CC/CA/AA; for IL-18 − 137G/C, GG/GC/CC; for IL-18 − 656C/A, CC/CA/AA; for IL-18 − 119A/C, AA/AC/CC; for IL-18 − 105G/A, GG/GA/AA.

**Table 7 ijms-24-00132-t007:** Studies for IL-6 gene polymorphisms and RPL.

Gene Polymorphism	Authors	Race	Number of Study and Control Groups	Genotype		Test Method	Conclusions
				Study Group	Control Group		
−174G/C	Unfried, G. et al., 2003 [240]	White Middle- European Caucasian women	161 vs. 124	66/72/23 ^a^	43/58/23 ^a^	PCR	NS
	Daher, S. et al., 2003 [219]	Brazilian	44 vs. 108	39/5 ^b^	99/9 ^b^	PCR	NS
	Prigoshin, N et al., 2004 [221]	Argentinean	38 vs. 54	35/3 ^b^	49/5 ^b^	PCR-SSP	NS
	Saijo, Y. et al., 2004 [241]	Japanese	76 vs. 93	76/0 ^c^	93/0 ^c^	PCR	NS
	Demirturk, F. et al., 2014 [242]	Turkish	113 vs. 113	72/36/5 ^a^	100/11/2 ^a^	PCR-RFLP	−174G/C polymorphisms had a relationship with an elevated incidence of RPL.
	Ma, J. et al., 2017 [207]	Chinese	775 vs. 805	484/248/43 ^a^	463/291/51 ^a^	PCR- RFLP	NS
−634C/G	Saijo, Y. et al., 2004 [241]	Japanese	76 vs. 93	58/18 ^c^	56/37 ^c^	PCR	Women with the G allele were more likely to develop RPL than those with the wild-type allele C.
	Ma, X. et al., 2012 [243]	Chinese	162 vs. 156	116/46/0 ^a^	93/52/11 ^a^	PCR-RFLP	The distributions of the GG genotype and G allele were significantly decreased in RPL.
	Ma, J. et al., 2017 [207]	Chinese	775 vs. 805	554/197/24 ^a^	478/277/50 ^a^	PCR- RFLP	Women with CG and GG genotypes were less likely to develop RPL.
−572G/C	Demirturk, F. et al., 2014 [242]	Turkish	113 vs. 113	81/28/4 ^a^	88/21/4 ^a^	PCR-RFLP	NS
−597G/A	Demirturk, F. et al., 2014 [242]	Turkish	113 vs. 113	96/16/1 ^a^	87/26/0 ^a^	PCR-RFLP	NS
−1363G/T	Demirturk, F. et al., 2014 [242]	Turkish	113 vs. 113	95/18/0 ^a^	94/19/0 ^a^	PCR-RFLP	NS
−2954G/C	Demirturk, F. et al., 2014 [242]	Turkish	113 vs. 113	107/6/0 ^a^	112/1/0 ^a^	PCR-RFLP	−2954G/C polymorphism variants were related to an elevated risk of RPL.

^a^ Genotype for IL-6 − 174G/C, GG/GC/CC; for IL-6 − 634C/G, CC/CG/GG; for IL-6 − 572G/C, GG/GC/CC; for IL-6 − 597G/A, GG/GA/AA; for IL-6 − 1363G/T, GG/GT/TT; for IL-6 − 2954G/C, GG/GC/CC; ^b^ Genotype for IL-6 − 174G/C, GG+GC/CC; ^c^ Genotype for IL-6 − 174G/C, GG/GC+CC; for IL-6 − 634C/G, CC/CG+GG.

**Table 8 ijms-24-00132-t008:** Studies for IL-1β, IL-1RA, and IL-1RN gene polymorphisms and RPL.

Gene Polymorphism	Authors	Race	Number of Study and Control Groups	Genotype		Test Method	Conclusions
				Study Group	Control Group		
IL-1β + 3954C→T on exon 5	Hefler, L. et al., 2001 [83]	Caucasian	131 vs. 68	79/46/6 ^a^	47/16/5 ^a^	PCR	NS
	Reid, J. et al., 2001 [244]	British	17 vs. 43	11/6/0 ^b^	26/14/3 ^b^	PCR	NS
	Ma, X. et al., 2012 [243]	Chinese	162 vs. 156	124/38/0 ^c^	130/26/0 ^c^	PCR-RFLP	NS
	Ma, J. et al., 2017 [207]	Chinese	775 vs. 805	602/168/5 ^c^	632/166/7 ^c^	PCR-RFLP	NS
IL-1β − 511C/T	Hefler, L. et al., 2002 [245]	Caucasian	130 vs. 67	29/90/11 ^c^	20/38/9 ^c^	PCR	NS
	Linjawi, S. et al., 2005 [246]	British	206 vs. 224	69/117/20 ^c^	85/110/29 ^c^	PCR	NS
	Ma, X. et al., 2012 [243]	Chinese	162 vs. 156	38/84/40 ^c^	46/84/26 ^c^	PCR-RFLP	NS
	Kim, J. et al., 2014 [232]	Korean	385 vs. 232	96/190/99 ^c^	39/120/73 ^c^	PCR–RFLP	−511C > T polymorphism was relevant to RPL.
	Ma, J. et al., 2017 [207]	Chinese	775 vs. 805	178/384/213 ^c^	156/392/257 ^c^	PCR-RFLP	NS
IL-1 RA	Unfried, G. et al., 2001 [247]	Austrian	105 vs. 91	0.9/0.34/0.5 ^d^	0.97/0.11/0.5 ^d^	PCR	Allele 2 was a genetic risk factor for RPL.
IL-1 RN	Wang, Z. et al., 2002 [248]	Caucasian	118 vs. 60	88/26/2/2 ^e^	39/20/1/0 ^e^	PCR-RFLP	NS
	Karhukorpi, J. et al., 2003 [249]	Finnish	37 vs. 800	12/19/2/4 ^f^	374/343/66/17 ^f^	PCR	The frequency of the IL-1RN*3 allele was significantly increased in RPL patients.
	Linjawi, S. et al., 2005 [246]	British	206 vs. 259	12/79/115 ^g^	17/92/150 ^g^	PCR	NS

^a^ Genotype for IL-1β + 3954, E1/E1: E1/E2: E2/E2; ^b^ Genotype for IL-1β + 3954, 1, 1/1, 2/2, 2; ^c^ Genotype for IL-1β + 3954, CC/CT/TT; for IL-1β − 511C/T, CC/CT/TT; ^d^ Genotype for IL-1 RA, 1/2/3; ^e^ Genotype for IL-1 RN, 1/2/3/4; ^f^ Genotype for IL-1 RN, 1/1: 1/2: 2/2: 1/3 or 2/3; ^g^ Genotype for IL-1 RN, 2, 2: 2, 4: 4, 4.

## Data Availability

Not applicable.

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
