# Peer review of "The Update Immune-Regulatory Role of Pro- and Anti-Inflammatory Cytokines in Recurrent Pregnancy Losses"

_ijms, 2022, doi:10.3390/ijms24010132_

Round 1

Reviewer 1 Report

To the autors

I have read your manuscript entitled “The update immune-regulatory role of pro- and anti-inflammatory cytokines in recurrent pregnancy losses

An underlying complex immune dysregulation appears to be one of the main factors that contributes to the maternal-foetal rejection and ultimately provoking the implantation failure or miscarriage. However, although animal models go in favour of this theory, human data are scarce and, unfortunately, disputed. All of this is especially true when we refer to the cytokine levels and particularly their balance.

Thus, the communication of the updated data on this topic is timely and welcome.

I would like to suggest some comments that, in my opinion, may improve this well-written manuscript.

Minor comments

Page 2: Introduction. Include subheading or paragraph on regulatory uNK cells

IL-10 must be mentioned and referred as an Th2-derived cytokine

I suggest including a specific paragraph on the complex: tolerogeneic dendritic cells, Tregs FoxP3+, IL-10 and embryo-paternal derived antigens. This, could be also linked to a more detailed role played by HLA-C genotype and its corresponding NK-KIR receptors.

Since different cytokines may by themselves exert anti-and pro-implantatory properties, I think could be interesting to build a Table depicting these capabilities. This,  may facilitate the understanding by the readers.

Figure 1: Il-10 should be included

Figure 2. It is difficult to understand. I would suggest to modify it. (To make it easier to understand)

Author Response

Comment 1: Page 2: Introduction. Include subheading or paragraph on regulatory uNK cells.

Response 1: Thank you for your comment. We have added the following sentences in lines 57-62:

“Regulatory uterine NK (uNK) cells refer to highly granulated NK cells located in the endometrium with the phenotype of CD56++/CD16- [16]. They have the ability to secrete a range of cytokines [16]. In normal pregnancy, increased regulatory uNK cells are pivotal for maintaining reproductive success, since they play a fundamental role in trophoblast invasion, spiral artery remodeling, and appropriate placentation [17-19]. Decreased number or dysfunction of regulatory uNK cells are closely related with RPL [20].”

Comment 2: IL-10 must be mentioned and referred as a Th2-derived cytokine.

Response 2: We revised the sentence in line 84:

“Th2 cytokines are predominantly IL-4, IL-5, IL-10, and IL-13 [24, 25].”

Comment 3: I suggest including a specific paragraph on the complex: tolerogeneic dendritic cells, Tregs FoxP3+, IL-10 and embryo-paternal derived antigens. This, could be also linked to a more detailed role played by HLA-C genotype and its corresponding NK-KIR receptors.

Response 3: We have added the following sentences in lines 63-74:

“The interaction between tolerogenic dendritic cells (DCs) and Treg cells in pregnancy is pivotal for sustaining maternal-fetal immune tolerance. Tolerogenic DCs promote the proliferation and differentiation of Treg cells by increasing the expression of IL-10 [21]. These special Treg subsets play a crucial role in avoiding maternal immune response against embryo-paternal derived antigens. Furthermore, tolerogenic DCs interacting with trophoblasts could elevate the proportion of Tregs FoxP3+ [22]. However, this phenomenon does not exist in DCs that do not interact with trophoblast cells [22]. Similarly, NK cells express killer cell immunoglobulin-like receptors (KIRs), which interact with HLA-C derived from extravillous trophoblast cells (EVTs). The improper combination between KIR and HLA-C affects the ability of NK cells to secrete cytokines including granulocyte-macrophage colony-stimulating factor (GM-CSF), thus participating in the occurrence of RPL [4].”

Comment 4: Since different cytokines may by themselves exert anti-and pro-implantatory properties, I think could be interesting to build a Table depicting these capabilities. This, may facilitate the understanding by the readers.

Response 4: We built Table 1 in line 692:

Table 1. Cytokines exert anti- or pro-implantatory properties.

Anti-implantation

IL-6

[234]

IL-17

[107]

IL-23

[107]

Pro-implantation

TNF-α

[131]

IFN-γ

[28]

TGF-β1

[235]

IL-1

[236]

IL-1β

[68]

IL-10

[237]

IL-11

[238]

IL-22

[121]

IL-33

[84]

Comment 5: Figure 1: IL-10 should be included

Response 5: We have added IL-10 in new Figure 1.

Comment 6: Figure 2. It is difficult to understand. I would suggest to modify it. (To make it easier to understand)

Response 6: We have separated Figure 2 into two figures (new Figure 2 and Figure 3) in the revised manuscript.

Reviewer 2 Report

This very detailed and complete manuscript describes what is known about the immunoregulatory cytokines in recurrent pregnancy losses (RPL).  I  think it is a little over done. As many of the factors discussed have discordant results of their role in RPL, they can likely be left out of this already cumbersome manuscript (for example section 2.6 and much of section 3.3). Manya of the sections describe published literature that is too preliminary and can be left out (for example section 2.8).  I would include only those with documented roles and consistent findings as to increased or decreased in RPL. I also recommend tables to summarize the data. A table on factors in the maternal peripheral blood and another table for endometrial/decidual factors and whether they are upregulated or downregulated would be extremely helpful. I find Figure 2 very confusing and unhelpful. Perhaps breaking that figure into at least two figures would clarify what the authors are trying to summarize.  The authors need to soften their summary statements especially when the data is contradictory to say "may have: or "likely play" a role in RPL or maintaining a viable pregnancy.
Other suggestions/comments:

Line 226: what are the decidual interstitial cells?

Lines 228-229: I could not find the aforementioned sequential changes, specify please.

Puy a blank line between the last sections and the summary paragraph.

How does race play a role in immunoregulatory cytokines in pregnancy (for example statement lines 415-416)?

There are English errors throughout, easily fixed I think by a native English speaking scientist.

Author Response

Comment 1: This very detailed and complete manuscript describes what is known about the immunoregulatory cytokines in recurrent pregnancy losses (RPL). I think it is a little over done. As many of the factors discussed have discordant results of their role in RPL, they can likely be left out of this already cumbersome manuscript (for example section 2.6 and much of section 3.3). Many of the sections describe published literature that is too preliminary and can be left out (for example section 2.8). I would include only those with documented roles and consistent findings as to increased or decreased in RPL.I also recommend tables to summarize the data. A table on factors in the maternal peripheral blood and another table for endometrial/decidual factors and whether they are upregulated or downregulated would be extremely helpful. I find Figure 2 very confusing and unhelpful. Perhaps breaking that figure into at least two figures would clarify what the authors are trying to summarize.  The authors need to soften their summary statements especially when the data is contradictory to say "may have: or "likely play" a role in RPL or maintaining a viable pregnancy.

Response 1: Thank you for your comment. We have deleted the original sections 2.6, 2.8, and much of section 3.3. We have built Table 2 on cytokines in the maternal peripheral blood in line 694 and Table 3 for endometrial/decidual cytokines in line 696. We have separated original Figure 2 into two figures (new Figure 2 and Figure 3) in the revised manuscript. We have revised the following sentences in lines 27-28 to soften the summary statements:

“Genetic polymorphism may be the underlying etiologies for Th1 and Th17 propensity since it alters cytokine production.”

Lines 34-36 and 106-107:

“Understanding the roles of cytokines involved in RPL might significantly advance the early diagnosis, monitoring, and treatment of RPL.”

Lines 126-127:

“Th1/Th2 ratios may be utilized to predict pregnancy outcomes, closely monitor pregnancy, and swiftly treat patients at risk of miscarriage.”

Lines 139-142:

“Therefore, a complicated cytokine network may determine the balance between Th1/Th2 immune responses during pregnancy with a propensity toward Th2 immune response after implantation is over, which in turn, again biased to Th1 immune response at the time of parturition [7].”

Lines 320-323:

“Significantly elevated expressions of IL-7 in the decidua of women with spontaneous abortion and RPL indicate that IL-7 likely plays a significant role in promoting pro-inflammatory immune response at the embryo-maternal interface [109].”

Lines 452-453:

“Therefore, RPL might be potentially addressed by correcting IL-11 signaling deficiencies in the endometrium [157].”

Lines 500-501:

“In RPL patients, NK cytotoxicity may be utilized as a biomarker to predict future abortions [175, 176].”

Lines 561-565:

“In addition, gene polymorphisms may affect the expression levels of cytokines, as well as their functions, which play an important role in the pathogenesis of RPL [202]. Therefore, cytokine gene polymorphisms might be utilized to predict abortion, stratify pregnancy management, and improve maternal and fetal pregnancy outcomes by increasing the surveillance of the high-risk abortion group.”

Lines 664-666:

“Consequently, understanding the intricate mechanisms of various cytokines may aid in establishing therapeutic modalities for RPL of immune etiologies and achieving a successful pregnancy.”

Lines 679-682:

“In women with RPL and dysregulated cytokine profiles, local or systemic administration of cytokines or their antagonists may be a potential therapeutic approach for maintaining immunological balance and improving maternal and fetal outcomes.”

Comment 2: Line 226: what are the decidual interstitial cells?

Response 2: We have revised the following sentence in lines 251-253:

“These activated pathways upregulate the CCL2/CCR2 axis, subsequently enhancing the proliferation and invasion of decidual stromal cells [85].”

Comment 3: Lines 228-229: I could not find the aforementioned sequential changes, specify please.

Response 3: We have revised the following sentence in lines 254-255:

“The reduced serum IL-33 level in RPL may be attributable to the down-regulated CCL2.”

Comment 4: Puy a blank line between the last sections and the summary paragraph.

Response 4: We have added a blank line between the last sections and the summary paragraph.

Comment 5: How does race play a role in immunoregulatory cytokines in pregnancy (for example statement lines 415-416)?

Response 5: We have revised the following sentence in lines 431-432:

“The disparities in these studies may be attributable to the heterogeneity of the different populations and the various underlying disorders.”

Comment 6: There are English errors throughout, easily fixed I think by a native English speaking scientist.

Response 6: Thank you for your suggestion. Recently, this manuscript has been edited by the expert staff and native English speaking scientist from a company. Changes have been marked in red in the revised manuscript.

Reviewer 3 Report

Manuscript ID: ijms-1999031

Title: The update immune-regulatory role of pro- and anti-inflammatory cytokines in recurrent pregnancy losses

Authors: Xiuhua Yang, Yingying Tian , Linlin Zheng , Thanh Luu , Joanne Kwak-Kim

This is a very interesting, updated review of the role of pro- and anti-inflammatory cytokines in recurrent pregnancy losses. While acknowledging the complexity of the immune network and the multifactorial nature of the treatment responses, the authors concluded that assessing a panel of cytokines and cytokine genes is likely to be necessary for assessing women with RPL. While this reviewer agrees with most of the arguments presented in this review manuscript, the author's brief account of the quintessential roles of the female major sex steroid hormones estrogen (E2) and progesterone (P4) would have to be revisited with more evidence to support their current conclusions on the anti-inflammatory role of E2 in pregnancy and the indications for the use of P4 in treating RPL.

Main concerns:

1- As communicated above, the main concern of this reviewer is the authors’ elaborated account of the systemic and local endometrial cytokine imbalances in women with RPL while providing scanty details on the role of the estrogen and progesterone in the pathophysiology of RPL (e.g., lines 609- 627 of the msms). Moreover, the authors made major conclusions that estrogen, a known pro-inflammatory sex steroid can regulate local cytokine balance in favor of Th2 type immune response without providing sufficient evidence to support their claims (e.g., the authors argued in favor of one solitary reference (AbdulHussain et al 2020)). It would be advantageous should the authors provide further evidence and elaborated on the contentious role of estrogen as an anti-inflammatory mediator during implantation and placentation conducive to providing a comprehensive analysis of the immune and molecular mechanisms regulated by E2 in the maternal decidua during the establishment and maintenance of early human pregnancy. Please reconsider updating your arguments in lines 609- 627 accordingly.

2- Please try to limit self-citations. The authors self-cited themselves ~ 10 times throughout the manuscript. Please diversify your citations accordingly. 

Author Response

Comment 1: This is a very interesting, updated review of the role of pro- and anti-inflammatory cytokines in recurrent pregnancy losses. While acknowledging the complexity of the immune network and the multifactorial nature of the treatment responses, the authors concluded that assessing a panel of cytokines and cytokine genes is likely to be necessary for assessing women with RPL. While this reviewer agrees with most of the arguments presented in this review manuscript, the author's brief account of the quintessential roles of the female major sex steroid hormones estrogen (E2) and progesterone (P4) would have to be revisited with more evidence to support their current conclusions on the anti-inflammatory role of E2 in pregnancy and the indications for the use of P4 progesterone in treating RPL.

As communicated above, the main concern of this reviewer is the authors’ elaborated account of the systemic and local endometrial cytokine imbalances in women with RPL while providing scanty details on the role of the estrogen and progesterone in the pathophysiology of RPL (e.g., lines 609- 627 of the msms). Moreover, the authors made major conclusions that estrogen, a known pro-inflammatory sex steroid can regulate local cytokine balance in favor of Th2 type immune response without providing sufficient evidence to support their claims (e.g., the authors argued in favor of one solitary reference (AbdulHussain et al 2020)). It would be advantageous should the authors provide further evidence and elaborated on the contentious role of estrogen as an anti-inflammatory mediator during implantation and placentation conducive to providing a comprehensive analysis of the immune and molecular mechanisms regulated by E2 in the maternal decidua during the establishment and maintenance of early human pregnancy. Please reconsider updating your arguments in lines 609- 627 accordingly.

Response 1: Thank you for your comment. We have added the following sentences in lines 622-624 about the anti-inflammatory role of E2 in pregnancy:

“The decrease of estrogen (E2) and progesterone (P4) induced the production of Th1 pro-inflammatory cytokines [220]. In addition, E2 and P4 enhance Th2 anti-inflammatory response and reduce the activity of pro-inflammatory macrophages and NK cells [221].”

Lines 648-651 about the contentious role of E2:

“However, E2 administration significantly reduced the circulating IL-17 expression from allergic encephalitis mice [230]. These discrepant results imply that it is necessary to explore the effect of E2 on Th17 cytokines in pregnant women.”

We have added the following sentences in lines 629-641 about the indications for the use of P4 progesterone in treating RPL:

“A few studies investigated the indications for the use of P4 in treating RPL. A large sample study involving 826 subjects revealed that compared with the placebo group, vaginal micronized progesterone plays a weak or no difference in improving the living birth rate of RPL patients [226]. Similarly, the proof of dydrogesterone for treating RPL is low certainty compared to placebo [226]. There is no sufficient data to analyze the effect of 17-α-hydroxyprogesterone or oral micronized progesterone on increasing the living birth rate of RPL patients [226]. In another study, the subgroup analysis of three experiments contained patients with three or more recurrent miscarriages [227]. P4 significantly reduced the abortion rate compared with placebo or no therapy [227]. It seems that P4 has a significant therapeutic effect on RPL. Therefore, they pointed out that P4 could be considered for treating RPL women, since the abortion rate was lower and there was no augment in side effects of mothers and neonates [227].”

Comment 2: Please try to limit self-citations. The authors self-cited themselves ~ 10 times throughout the manuscript. Please diversify your citations accordingly. 

Response 2: We deleted the following four self-citations in the revised manuscript:

Yang X, Yang E, Wang WJ, He Q, Jubiz G, Katukurundage D, et al. Decreased Hla-C1 Alleles in Couples of Kir2dl2 Positive Women with Recurrent Pregnancy Loss. Journal of reproductive immunology (2020) 142:103186. Epub 2020/08/28. doi: 10.1016/j.jri.2020.103186.

Liu S, Diao L, Huang C, Li Y, Zeng Y, Kwak-Kim JYH. The Role of Decidual Immune Cells on Human Pregnancy. Journal of reproductive immunology (2017) 124:44-53. Epub 2017/10/23. doi: 10.1016/j.jri.2017.10.045.

Ntrivalas EI, Kwak-Kim JY, Gilman-Sachs A, Chung-Bang H, Ng SC, Beaman KD, et al. Status of Peripheral Blood Natural Killer Cells in Women with Recurrent Spontaneous Abortions and Infertility of Unknown Aetiology. Human reproduction (Oxford, England) (2001) 16(5):855-61. Epub 2001/05/02. doi: 10.1093/humrep/16.5.855.

Choi YK, Kwak-Kim J. Cytokine Gene Polymorphisms in Recurrent Spontaneous Abortions: A Comprehensive Review. American journal of reproductive immunology (New York, NY: 1989) (2008) 60(2):91-110. Epub 2008/06/25. doi: 10.1111/j.1600-0897.2008.00602.x.

Round 2

Reviewer 2 Report

Thank you for your careful responses to my suggestions.

Reviewer 3 Report

Thank you for your revisions.